# Allergic Asthma in the Era of Personalized Medicine

**DOI:** 10.3390/jpm12071162

**Published:** 2022-07-18

**Authors:** Niki Papapostolou, Michael Makris

**Affiliations:** Allergy Unit, 2nd Department of Dermatology and Venereology, Medical School, National and Kapodistrian University of Athens, “Attikon” University Hospital, 12462 Athens, Greece; nikipapap@gmail.com

**Keywords:** allergic asthma, asthma, allergic phenotype, allergen-specific immunotherapy, biomarkers

## Abstract

Allergic asthma is the most common asthma phenotype and is characterized by IgE sensitization to airborne allergens and subsequent typical asthmatic symptoms after exposure. A form of type 2 (T2) airway inflammation underlies allergic asthma. It usually arises in childhood and is accompanied by multimorbidity presenting with the occurrence of other atopic diseases, such as atopic dermatitis and allergic rhinitis. Diagnosis of the allergic endotype is based on in vivo (skin prick tests) and/or in vitro (allergen-specific IgE levels, component-resolved diagnosis (CRD)) documentation of allergic sensitization. Biomarkers identifying patients with allergic asthma include total immunoglobulin E (IgE) levels, fractional exhaled nitric oxide (FeNO) and serum eosinophil counts. The treatment of allergic asthma is a complex procedure and requires a patient-tailored approach. Besides environmental control involving allergen avoidance measurements and cornerstone pharmacological interventions based on inhaled drugs, allergen-specific immunotherapy (AIT) and biologics are now at the forefront when it comes to personalized management of asthma. The current review aims to shed light on the distinct phenotype of allergic asthma, ranging over its current definition, clinical characteristics, pathophysiology and biomarkers, as well as its treatment options in the era of precision medicine.

## 1. Introduction

Asthma is a chronic heterogeneous disease affecting more than 330 million people worldwide [1,2]. It constitutes a significant socioeconomic burden, especially in its severe uncontrolled form [3]. Over the next 20 years, asthma management in the United States is predicted to cost more than $960 billion [4]. Asthma is usually characterized by chronic airway inflammation and variable symptoms including cough, chest tightness, shortness of breath and wheezing, along with variable expiratory airflow limitation, which can become persistent due to airway remodeling [5]. Based on age of onset, atopy status, preservation of airflow, comorbidities, exacerbations, response to treatment and prognosis, as well as the underlying airway inflammation, different phenotypes and endotypes are embraced by the definition of asthma [6]. Hence, “asthma” has been proposed as an umbrella term under which all these phenotypes and endotypes, frequently overlapping, lie, with allergic asthma holding a key position [7]. It is estimated that allergic mechanisms are present in about 80% of child asthmatics and 40–50% of adult asthmatics [8]. Thus, the allergic phenotype is the most common asthma phenotype in the general asthma population [9,10] and the most commonly reviewed allergic disease in the literature [11]. The frequency of allergic asthma and the difficulty in precisely recognizing and differentiating it from other type 2 (T2) overlapping endotypes in clinical practice constitute a great challenge in the era of the pursuit of therapeutic strategies made to measure for every patient; that is, the era of precision medicine and tailor-made interventions. The current review aims to provide an overview of data regarding the phenotype definition, clinical characteristics, pathophysiology, possible biomarkers and current therapeutic options for allergic -asthma (Table 1).

## 2. Definition

Since the nature of asthma is complex and heterogeneous, framing the definition of allergic asthma is a great challenge. An “allergy”—and, therefore, “allergic”—is currently defined as an immune-mediated hypersensitivity reaction [8]. Recently, however, a more provocative—albeit refreshing—definition has been proposed [12]. Allergic asthma is usually defined as asthma associated with sensitization to aeroallergens. This definition consists of two criteria: (A) identification of allergic sensitization (in vivo and/or in vitro) and (B) association between aeroallergen exposure and asthma symptoms [9]. However, allergic asthma is often defined in the literature only with regard to the presence of sensitizations independently of their correlation with symptoms, thus further blurring the picture of this phenotype. Moreover, severe asthma represents one of the main goals of precision medicine, with monoclonal antibodies being at the forefront. Hence, when choosing a biologic agent, a stricter definition of allergic asthma is required. As derived from the populations of the randomized controlled trials (RCTs) that informed the regulatory approval in current published guidelines regarding biologics for severe asthma, the allergic subtype is clearly distinguished from the eosinophilic one. It is defined as the asthma phenotype characterized by symptoms resulting from exposure to a perennial aeroallergen and serum total IgE levels of 30–1300 IU/mL that are not adequately controlled with high doses of inhaled corticosteroids/long-acting beta2-agonists (ICSs/LABAs) and/or other background controllers [13]. This definition seems to be of particular use in the selection of omalizumab-eligible patients within the context of personalized medicine for severe allergic asthma. However, to complicate the puzzle even more, despite the clear distinction between allergic and eosinophilic asthma stated in current guidelines, different treatment options are currently proposed that ignore or misconceive the overlapping natures of these subtypes in clinical practice [13,14].

## 3. Clinical Characteristics

Allergic asthma often starts in childhood [15,16], presenting with a large variation in disease severity [17,18], and it can occasionally be persistent and continue into adulthood [19,20,21]. Although individuals with childhood asthma are expected to outgrow their asthma before adulthood, in about 30% of children with asthma the condition persists [22,23]. Comorbidities associated with allergic asthma, such as allergic rhinitis and atopic dermatitis, have been associated with lower likelihood of asthma remission [21]. The Swedish OLIN cohort of patients with asthma demonstrated that allergic sensitization to furred animals was associated with lower probability of asthma remission in adulthood [24]. Recent results from the Tasmanian Asthma Study suggest that early-onset asthma and allergic phenotypes are associated with worse clinical outcomes later in life and a higher predisposition to chronic obstructive pulmonary disease (COPD) when compared to individuals with less allergic phenotypes [20]. Moreover, in the allergic asthma phenotype, family history of asthma and other atopic diseases is also present [10,16,25,26,27]. The occurrence of allergic comorbidities, such as allergic rhinitis or atopic dermatitis, is common, providing support for “multimorbidity” [26,28], a term that emerged to cluster allergic diseases with a tendency to coexist in individuals. These chronic allergic diseases share common immune-dysregulation and inflammation profiles on the basis of genetic polymorphisms [29].

Severity occurs in allergic asthma with great heterogeneity. Results from the Epidemiological Study on the Genetics and Environment of Asthma (EGEA) supported the idea that sensitization to environmental allergens is common in asthmatic children (88.2%), but there was no association found between asthma severity and atopy [17]. However, Fitzpatrick et al. found that allergic sensitization in school children with asthma was more prevalent in severe than in mild to moderate asthmatics [18]. Further epidemiological studies support the idea that atopy and sensitization to environmental allergens are associated with severe pediatric asthma [30,31]. More recently, Durfois et al., evaluating the maintenance of asthma control during the transition to adult centers in a cohort of adolescents with severe asthma, reported that 80% had allergic asthma and allergic multimorbidities and were partially controlled during follow-up [32]. In an attempt to characterize severe asthma worldwide, additional data from the International Severe Asthma Registry showed that 50% of patients with severe asthma had high IgE levels (>150) and that IgE levels varied according to severity; patients receiving Global Initiative for Asthma step-five treatment had higher total IgE levels compared to those receiving step-four treatment [33]. Moreover, in a prospective study of 50 severe asthmatic female patients, near-fatal asthma exacerbations were more likely in cases of type 2-high (T2-high) atopic asthma [34].

Allergic asthma in childhood may progress into severe asthma in adulthood [16,19]. In a birth cohort study following children up to 13 years of age prospectively, early wheezing episodes and sensitization to perennial environmental allergens were associated with asthma persistence during puberty [35].

Viral respiratory infections early in childhood, and especially rhinovirus (RV) infections, are implicated in asthma development and acute exacerbations, although their role in asthma persistence is debatable [36,37]. On the other hand, sensitization to environmental allergens in early life is strongly associated with asthma persistence, while the relationship it has with asthma initiation is still not well-documented [38]. Thus, there are growing implications that factors contributing to the initiation and persistence of asthma are distinct and that complex, dynamic interactions between environmental exposure and genetic susceptibility underlie the clinical presentation of the allergic asthma phenotype [38,39]. Cluster analysis using omics technologies along with gene expression data have been used to describe different asthma phenotypes. Allergic asthma constitutes a common cluster in many large-scale multicenter trials (U-BIOPRED, SARP, UK clusters) [16,40,41]. Recently, a review has suggested four 4 distinct asthma phenotypes can be identified according to the abovementioned studies: (a) early-onset allergic asthma, (b) early-onset moderate to severe allergic asthma with a reduced forced expiratory volume in 1 s (FEV1), (c) late-onset non-allergic eosinophilic asthma and (d) late-onset non-allergic non-eosinophilic asthma [42]. However, proposing asthma endotypes using cluster analysis carries numerous limitations, as various external factors could affect the analysis and no data replication is present [16,40,41].

## 4. Pathogenic Mechanisms in Allergic Asthma

Asthma is a heterogeneous, well-defined syndrome. A complex interplay between environmental and genetic factors along with underlying inflammation leads to the expression of distinct clinical phenotypes. Hence, the underlying pathogenic mechanisms are quite intricate and vary among asthmatics [16].

The modern concept of classification of immune dysregulation in asthma has led to the description of asthma type 2-high inflammation (T2-high, eosinophilic) and type 2-low inflammation (T2-low, non-eosinophilic) endotypes. (Figure 1) T2-high asthma is characterized by eosinophilic airway inflammation with high blood eosinophil counts and elevated levels of fractional exhaled nitric oxide (FeNO) [43]. Allergens or other triggers can initiate this eosinophilic inflammation through innate or adaptive immunity and, thus, the term “T-helper 2-high” (TH2-high) inflammation has been replaced by the term “type 2-high” (T2) to encompass innate lymphoid cells type 2 (ILC2s) and T-helper type 2 (TH2), associated with asthma inflammation [44]. In contrast, type 2-low asthma embraces neutrophilic and paucigranulocytic asthma. The coexistence of eosinophilic and neutrophilic airway inflammation characterizes the mixed granulocytic endotype [43].

Allergic asthma, lying under the umbrella of the T2-high endotype, is characterized by eosinophilic airway inflammation and is the most well-defined asthma model.

The role of the epithelial barrier in allergic diseases and other chronic diseases is beyond question [45]. Activation or dysregulation of epithelial cells is one of the initiators of inflammation in both T2-high and in T2-low asthma. Once disruption of the epithelial barrier constitutes “leakage” and multiple immune-regulatory mechanisms arise to suppress the inflammation, dynamic interactions take place and, in susceptible individuals, multiple inflammatory diseases can arise [45].

Focusing on allergic asthma, a susceptible individual’s exposure to aeroallergens leads to activation of the epithelium in the airways and subsequent secretion of the epithelial-derived cytokines known as ”alarm signs” or ”alarmins”: interleukin 25 (IL-25), interleukin 33 (IL-33) and thymic stromal lymphopoietin (TSLP) [46]. Antigen-presenting cells present allergens to naïve T lymphocytes in local lymph nodes and, together with co-stimulatory molecules, naïve T cells differentiate into TH2 cells, secreting great amounts of type 2 cytokines: interleukin 4 (IL-4), interleukin 5 (IL-5) and interleukin 13 (IL-13) [44]. IL-4 plays a substantial role in the differentiation of T lymphocytes into the T-helper 2 subtype (TH2). Moreover, IL-4 and, to a lesser extent, IL-13 drive IgE isotype switching in B lymphocytes and, thus, secretion of IgE allergen-specific antibodies in response to allergen stimulation [47]. Those specific IgE antibodies are bound to high-affinity IgE receptors (FcεRI) on the surface of mast cells (MCs) and basophils (sensitization phase). In subsequent allergen exposure, myeloid dendritic cells in allergen-sensitized patients, together with co-stimulatory molecules, present inhaled aeroallergens to tissue-resident memory CD4+ type 2 helper lymphocytes (TH2) expressing allergen-specific T-cell receptors (TCRs), thus responding in an allergen-specific manner on activation. Aeroallergens cross-link IgE receptors on the surface of mast cells and basophils, leading to rapid and massive activation [48]. Degranulation of preformed mediators, such as histamine, tryptase, serotonin, proteases, proteoglycans, carboxypeptidase A and chymase, together with the formation of newly synthetized lipid mediators, such as prostaglandins and cysteinyl-leukotrienes, results in strong bronchoconstriction, mucus hypersecretion [49,50] and edema of the airway wall [51]. In addition to these acute responses, cytokines are produced from TH2 cells, mast cells and basophils, resulting in a late-phase reaction that is characterized by infiltration of inflammatory cells contributing to innate and adaptive immunity: eosinophils, basophils, CD4+ T-helper cells, memory cells, monocytes and neutrophils [52].

Moreover, IL-13 has a multifunctional role, as it has been associated with mucus hypersecretion, goblet-cell hyperplasia, subepithelial fibrosis, airway hyperreactivity and stimulation of nitric oxide synthase enzyme in bronchial epithelial cells, leading to increased levels of FeNO [52,53]. IL-5 orchestrates the proliferation, maturation, differentiation and survival of eosinophils. It also leads them to activate and secrete their mediators, resulting in tissue damage and inflammation [54]. Eosinophil numbers in blood, airway biopsy specimens and bronchoalveolar lavage fluid strongly correlate with the severity of asthma and exacerbations [52,55].

Furthermore, the innate immune system plays a central role in asthma pathogenesis through the innate lymphoid type 2 cells (ILC2) [56]. These cells secrete excessive amounts of T2 cytokines, such as IL-13 and IL-5, but not IL-4. Although they lack antigen-specific receptors, they express receptors such as chemoattractant receptor-homologous molecule expressed on TH2 cells (CRTh2), interleukin-7 receptor alpha chain (CD127) and interleukin-1 receptor like-1 (ST-2) and are activated by epithelial-derived cytokines. Moreover, those cells express high levels of GATA3 and, thus, contribute to TH2 activation and subsequent release of type 2 cytokines [57]. The pathway of ILC2s activation is associated with the eosinophilic, non-allergic, T2 endotype [58]. Secretion of IL-5 and IL-13 from ILC2s leads to eosinophilic inflammation but not IgE class switching, which is primarily driven by IL-4. Hence, IgE-driven mechanisms are associated with allergic asthma, while ILC2s-driven mechanisms are associated with eosinophilic asthma, with eosinophils per se contributing to both types of T2 asthma [59,60,61].

The persistence of allergic asthma and the natural course of remodeling is another significantly unexplored area. As already mentioned above, factors that initiate or exacerbate asthma may differ from those linked to disease persistence [39]. Allergen sensitization, especially in early childhood, has been linked with asthma persistence [38]. Viral infections. and especially rhinovirus infections (RV). have been linked to asthma initiation and exacerbation, as they promote airway remodeling and prolonged hyperresponsiveness [62,63,64]. Moreover, atopic patients have a predisposition for a T2-driven response and an impaired interferon T-helper 1 cell (TH1) response. Thus, exposure to both allergens and viruses, especially in early life, can result in airway inflammation and subsequent damage to airway tissues. Consequently, a patient exposed to both allergens and viral infections early on has a high risk of asthma persistence [19]. In line with this hypothesis, wheezing and atopy in the first year of life were independently associated with an increased risk of asthma by the age of six and the risk was even higher in children with both atopy and respiratory infections [65].

## 5. Biomarkers

A biomarker can be defined as any objectively measured characteristic that can be used as an indicator of disease diagnosis, a predictor of response to treatment and for monitoring [66]. To be applicable in clinical practice, a biomarker should be able to define a phenotype and/or an endotype, predict and evaluate treatment response and monitor disease progression [67]. As we enter the era of personalized medicine, identifying the perfect biomarker with the abovementioned characteristics is urgently needed, especially in allergic and other T2-overlapping severe asthma endotypes, for which the choice of the ideal treatment for each patient is quite challenging [68]. Although several biomarkers have been proposed to be associated with T2 inflammation, allergic asthma often presents with elevated levels of immunoglobulin E (IgE), FeNO, blood and sputum eosinophils and periostin [68].

### 5.1. Total and Specific Immunoglobulin E (IgE)

Measurement of total IgE levels and sensitization rates to environmental allergens, either through measurement of specific IgEs in serum and/or by in vivo performance of skin tests, have been used to identify patients with allergic asthma. A relationship between total IgE levels and allergic asthma has been shown [17]. Although no association with asthma severity was shown, IgE levels were associated with hospitalization rates and use of inhaled corticosteroids (ICS) in the Epidemiological Study on the Genetics and Environment of Asthma (EGEA) study [17]. In another study by Fitzpatrick et al., IgE levels were positively correlated with asthma severity [18]. Although the inextricable link between atopy and asthma is beyond question, it is well-accepted that not all atopics will show a clinical presentation of their IgE sensitization and, thus, development of allergic asthma [69]. Hence, it is important to highlight that allergen sensitization alone does not necessarily mean a clinically relevant allergy and expression of allergic asthma. Measurements of specific IgEs and skin prick tests for whole allergen extracts cannot differentiate between asymptomatic and clinically relevant sensitization. However, component-resolved diagnosis has emerged as a possible diagnostic tool in asthma and as a way to measure biomarkers for symptomatic sensitization [70]. Identification of sensitization clusters based on measurements and interactions of allergen components have shown that interplay between pairs of specific allergen components is associated with increased risk of asthma [71]. Moreover, in mite-sensitized children, the immunoglobulin G/immunoglobulin E ratio was significantly lower in children with asthma compared to those without asthma [72].

### 5.2. Fractional Exhaled Nitric Oxide (FeNO)

FeNO is a non-invasive biomarker associated with eosinophilic airway inflammation that can easily be measured in preschoolers [73,74,75]. IL-13 stimulates nitric oxide synthase in bronchial epithelial cells and leads to a subsequent increase in FeNO levels [52]. A modest association between eosinophilic airway inflammation and FeNO levels has been previously shown [76]. The Global Initiative for Asthma (GINA) recommends FeNO levels above 25 parts per billion (ppb) as a marker of T2 inflammation when assessing severe asthmatic patients [14].

FeNO levels above 50 ppb and 35 ppb in adults and children, respectively, are associated with eosinophilic inflammation and response to ICS [77]. However, FeNO levels can be elevated in cases of allergic rhinitis, atopy, eosinophilic bronchitis and atopic dermatitis and reduced in smokers or during bronchoconstriction. Viral exposure can result either in reduced or increased levels of FeNO [74].

Results from the Severe Asthma Research Cohort indicate that elevated levels of FeNO were associated with early-onset asthma, atopy and airway hyperreactivity [73]. A recent systematic review and metanalysis showed that measurement of FeNO levels has high specificity for diagnosis of asthma, with higher cut-off values (>50 ppb) increasing specificity further [78].

### 5.3. Eosinophils

Sputum eosinophil is one of the most well-characterized biomarkers for the assessment of severe asthma. Due to limitations in obtaining samples, especially in pediatric populations, it is usually performed in centers specializing in severe asthma in adult populations [67]. Results from a systematic review and meta-analysis indicated that sputum eosinophilia was moderately correlated with blood eosinophils, FeNO levels and total serum IgE [76].

Although elevated levels of eosinophils usually suggest response to corticosteroid therapy, they may indicate poor adherence or incorrect technique in ICS therapy [79,80].

Due to a reduction in exacerbation rates when a sputum-guided management was followed in severe asthmatics [81], GINA and European Respiratory Society/American Thoracic Society (ERS/ATS) recommend sputum-guided therapeutic interventions for severe asthma populations in specialized centers [82,83].

Measuring blood eosinophils is an easy intervention, and blood eosinophil is widely used as an eligibility marker (different cut-offs in different countries) for anti-IL-5 monoclonal antibody selection and as a predictor of response [82]. However, use of systemic corticosteroids, age and concomitant atopic disease has to be considered when evaluating blood eosinophils [84].

In a cross-sectional study, blood eosinophils, sputum eosinophils and elevated levels of total IgE (>100 IU/mL) were significantly correlated with severe persistent asthma [85]. In a systematic review and meta-analysis, Korevaar et al. found only moderate accuracy regarding the use of blood eosinophils, IgE and FeNO in predicting airway eosinophilia. If these biomarkers are considered in isolation, they have low sensitivity and specificity in identifying airway inflammation compared to sputum eosinophils [76]. Higher levels of peripheral eosinophils and TH2-polarized responses have been identified in patients with allergic asthma compared to those with non-allergic asthma [86]. Blood eosinophil levels in patients with allergic asthma vary according to different studies, with elevated levels (>300 cells/μL) being present in about 50% of patients in RCT and real-life studies [87,88,89]. In the STELLAIR study, a retrospective real-life study assessing omalizumab effectiveness in patients with allergic asthma, blood eosinophil count was ≥300 cells·µL^−1^ in 52.1% of the adults and 73.8% of the pediatric population [87]. In another real-life yet retrospective study, blood eosinophils ranged between 0 and 2340 cells/μL in 711 patients at baseline, with 35.4% of patients having levels >300 cell/μL [88].

### 5.4. Serum Periostin

Periostin, a matricellular protein that is upregulated by T2 cytokines, IL-4 and IL-13 in bronchial epithelial cells, is involved in T2 inflammation and airway remodeling in asthma [90,91]. The ability to measure periostin in the serum of asthmatic patients makes it a useful biomarker in assessing T2 inflammation and remodeling [92,93]. Measurement of serum periostin correlates with other T2 biomarkers, such as IgE levels, serum eosinophils, eosinophilic cationic protein and airway hyperresponsiveness [94,95].

Finally, the potential use of periostin as a biomarker in pediatric populations is yet to be explored since, due to bone growth, baseline periostin levels are higher in children [96].

## 6. Treatment

Treating potentially modifiable risk factors to reduce exacerbations is a major goal in asthma; in cases of allergen exposure in sensitized patients, controller treatment and environmental-control measures involving allergen avoidance, allergen-specific immunotherapy (AIT) and biologics are interventions of major importance, for which step-up methods can be considered [82]. Current asthma management aims at symptom control and reduction of future risk, such as the risk of exacerbations, asthma-related mortality, persistent airflow limitation and side effects of treatment. Multiple therapeutic interventions are used in asthma management, including controller and reliever medications [82].

### 6.1. Therapeutic Interventions in Asthma: Standard of Care

Apart from the use of ICSs, which constitute the gold standard in maintenance treatment of asthma and biologic agents, many other treatment options are available for asthma management. Long-acting b2-agonists (LABAs) approved for use with asthma include formoterol, salmeterol and vilanterol, and the combination of ICSs and LABAs is proposed for adults and adolescents in step 3 of the GINA report. In response to “Single Maintenance and Reliever Therapy”, in which a single inhaler is used as a controller and reliever medication, GINA recommends the use of as-needed low doses of ICS/formoterol as the preferred reliever medication for all levels of asthma severity in adults and adolescents [82]. Leukotriene receptor antagonists (LTRAs) and 5-lipoxygenase inhibitors can be used in patients with mild asthma as an alternative to ICSs, an alternative to ICS dose increases or as an add-on to the use of LABAs. LAMAs may be considered as an add-on therapy, using a separate inhaler for patients aged >6 years or in a triple combination in patients aged >18 years if asthma remains uncontrolled despite medium- or high-dose ICS/LABA treatment. Theophylline and cromoglycate are controllers that are rarely used nowadays due to their increased side effects and inferior efficacy profiles compared to ICSs. The use of SABA is no longer considered the best choice in the form of single inhalers for patients with mild asthma and they are no longer the preferred reliever option for all other patients. The short-acting muscarinic antagonist ipratropium can be used in a hospital setting along with SABA in asthma flare-ups to improve exacerbation [82].

### 6.2. Environmental Interventions

Although allergen avoidance seems to be an appealing way to achieve better control in allergic asthma, most of the time trigger avoidance is impractical and burdensome for the patient [82]. Regarding indoor allergen avoidance and the impact it has on asthma, conflicting data exist. However, a number of studies support the idea that environmental control can significantly affect allergic asthma management. In an RCT, the use of insecticidal bait to eradicate cockroaches in houses of children with moderate to severe asthma was associated with better symptom control, fewer emergency visits associated with exacerbations and a higher FEV1 compared to children in the non-interventional group [97]. A systematic review assessing the effect of multicomponent environmental interventions concluded that an overall improvement in productivity and quality of life was observed in children and adolescents with asthma, but the results regarding the adult population were inconclusive [98]. Another study that used intervention measurements to control dust mite and cockroach allergens in houses of inner-city children showed a reduction in asthma-associated morbidity [99]. Moreover, the use of mite-impermeable covers in mite-sensitized children led to a reduction of asthma exacerbation, but no effect was observed from the use of oral corticosteroids [100].

To conclude, allergen avoidance is not recommended as a general practice for patients with allergic asthma. Although it can be beneficial in sensitized patients, it is complex and expensive. Moreover, a clearly defined population in which those measures would be more beneficial is lacking [82].

### 6.3. Corticosteroids

Use of inhaled corticosteroids (ICSs) remains the cornerstone therapeutic intervention in asthma [101]. However, due to different underlying pathomechanisms, a variable response to ICSs is observed [102]. Patients with allergic asthma are usually among the best responders to ICSs [2]. Szefler et al. reported that, among children with mild to moderate asthma, higher levels of blood eosinophils, total IgE, eosinophil cationic protein and FeNO, accompanied by the presence of sensitization to environmental allergens, were associated with a favorable response to inhaled fluticasone [103]. In a multicenter study of children at substantial risk for asthma, allergic sensitization was associated with a better response to ICSs [104]. In another study by Fitzpatrick et al., blood eosinophils above 300 cells/μL, along with sensitization to aeroallergens, were associated with a higher probability of response to daily ICS treatment [105]. Moreover, in another recent study seeking to identify biomarkers predicting responses to ICSs and/or long-acting muscarinic antagonists (LAMAs) in patients with uncontrolled mild asthma, levels of IgE and sensitization to aeroallergens were associated with a favorable ICS response (AUCs: 0.73 (95% CI, 0.58–0.87) and 0.69 (95% CI, 0.52–0.85), respectively, both *p* < 0.03) [106]. Assessing data from the Pediatric Asthma Controller Trial (PACT) study to identify phenotypic characteristics predicting different treatment responses to fluticasone and montelukast, Knuffman et al. found that total serum IgE levels >150 KU/L, blood eosinophils >5% and FeNO levels >25 ppb were associated with a favorable response to fluticasone compared to montelukast [107]. In a large cohort study of adult asthmatics monitored for over 20 years, prolonged treatment with ICSs (>8 years) in subjects sensitized to any of four common allergens (house dust mites (HDMs), cats, Cladosporium and timothy grass) was associated with an attenuated decline in FEV1, indicating that biomarkers of allergic sensitization can predict a more favorable response to prolonged ICS treatment [108].

### 6.4. Allergen Immunotherapy

Allergen immunotherapy (AIT), with a 100-year-long history of use for allergic diseases, aspires to be the only etiologically based therapy for allergic asthma and allergic rhinitis and works by inducing immunologic tolerance [109,110]. AIT has a disease-modifying effect, demonstrated by prevention of new allergen sensitizations and the ability to halt disease progression [111,112,113]. Hence, allergen immunotherapy is recommended for patients with allergic rhinitis and allergic asthma according to multiple guidelines [5,114,115]. AIT is recommended for patients with well-controlled allergic asthma, although it is not recommended for severe uncontrolled asthma, as lack of control of symptoms is considered a risk factor for severe adverse events during AIT [111].

Regarding the route of administration, subcutaneous immunotherapy (SCIT) or sublingual immunotherapy (SLIT) are two options [116]. According to the recently published European Academy of Allergy and Clinical Immunology (EAACI) guidelines for the use of house-dust-mite (HDM) immunotherapy as an add-on treatment for HDM-driven allergic asthma in children and adolescents, HDM SCIT is recommended both in children and adults with allergic asthma to reduce symptoms and medication needs. HDM SLIT tablets and drops are recommended for adults and children, respectively [114].

GINA recommends the use of HDM SLIT in adults and adolescents with allergic rhinitis and HDM sensitization as an add-on treatment when asthma symptoms persist despite treatment with a low/medium dose of ICS-containing medication (as recommended in step 2 of the GINA report), given that FEV1 is predicted to be above 70% [5].

Allergic asthma usually starts in childhood with sensitization to environmental allergens, and AIT has long been suggested as a disease-modifying therapy in children with AR to prevent the progression to allergic asthma [113].

The use of SCIT for patients with allergic asthma can result in reduction of symptoms and medication needs and can also improve quality of life and bronchial hyperresponsiveness, but there is a less clear effect on lung function [117,118,119].

Although SLIT and SCIT have been shown to reduce symptoms and medication use, SLIT has a more favorable profile. A combination of SCIT and omalizumab has been shown to prevent adverse events and increase the effectiveness of SCIT [120,121]. More recently, the simultaneous use of AIT and omalizumab has been proposed as both a bridge to safely administer SCIT and as an omalizumab-sparing agent [122].

### 6.5. Biologic Agents

In patients with severe uncontrolled asthma characterized by T2 inflammation, targeted therapies using biologic agents are available as add-on treatments to reduce the disease burden [13]. Five monoclonal antibodies are currently licensed by both the Food and Drug Administration (FDA) and the European Medicines Agency (EMA) as add-on treatments for severe uncontrolled asthma: benralizumab, dupilumab, mepolizumab, omalizumab and reslizumab. Recently, tezepelumab, an anti-TSLP biologic, has been granted approval by the FDA for use in the treatment of severe asthma in adults and adolescents aged 12 years and older.

Omalizumab, a humanized monoclonal antibody binding free IgE at the FcεRI binding site, was the first biologic agent to be approved for asthma treatment by the FDA almost 20 years ago. This anti-IgE monoclonal antibody targets the C3 domain of the Fc fragment of free IgE and inhibits IgE binding to its high- (FcεRI) and low-affinity (FcεRII) receptors on the surface of mast cells, basophils, B cells and dendritic cells. Thus, it reduces free IgE levels in serum, inhibits the activation of IgE receptor-bearing cells and downregulates both high- and low-affinity receptors on the surfaces of those cells. Moreover, it enhances the production of interferon alpha (IFN-α) from plasmacytoid dendritic cells (pDCs), demonstrating an antiviral effect [13,123].

Omalizumab is approved as an add-on treatment for subcutaneous administration in patients six years of age and older with moderate to severe persistent allergic asthma (FDA) or severe persistent allergic asthma (EMA) despite treatment with ICSs and in vivo and/or in vitro proof of sensitization to perennial aeroallergens. Baseline IgE levels and body weight are used to optimize the treatment dose and schedule of administration. However, total IgE levels do not predict the response to treatment [124].

Omalizumab has been shown to reduce asthma exacerbations, improve symptom control and quality of life and reduce the use of ICS. Inconclusive data regarding lung function derive from RCTs [125,126,127]. Post hoc analysis showed that reductions in asthma exacerbations were greater in patients with high FeNO, blood eosinophil and periostin levels [89]. In spite of that, two recent real-life studies, PROSPERO and STELLAIR, demonstrated a clinically meaningful improvement in lung function in an adolescent population. Regarding the biomarker status of patients, the same studies demonstrated that response to omalizumab treatment was irrespective of IgE and blood eosinophil levels [87,88].

Dupilumab, a fully human monoclonal antibody binding to the IL-4 receptor alpha subunit, shared by both IL-4 and IL-13 receptor complexes, is the first dual inhibitor of IL-13 and IL-4 signaling granted approval by the FDA for moderate to severe asthma in patients older than six years of age with eosinophilic or corticosteroid-dependent asthma and by the EMA for severe type 2 asthma in patients older than 12 years. In RCTs, dupilumab decreased asthma exacerbations and symptom control, improved lung function and showed a corticosteroid-sparing effect [128,129,130]. Reductions in exacerbations and improvement in FEV1 were even distinct in patients with higher baseline blood eosinophil (>150/μL) and FeNO (25 ppb) levels [128,131]. Dupilumab is also approved for the treatment of atopic dermatitis and chronic rhinosinusitis with nasal polyps and is an appealing therapeutic option for patients with asthma and these co-existing comorbidities [132,133]. Patients with blood eosinophils >1500/μL should be investigated for other hypereosinophilic conditions if indications are present and dupilumab should preferably be avoided, as those patients were excluded in phase III studies [134].

Mepolizumab, reslizumab and benralizumab are three monoclonal antibodies targeting IL-5 and the IL-5 receptor. They are approved by the FDA and the EMA for severe eosinophilic asthma. Mepolizumab is licensed for patients older than 6 years of age, benralizumab for patients above 12 years and reslizumab for the adult population. Both reslizumab and mepolizumab target IL-5, preventing it from binding to its receptor, while benralizumab binds to the IL-5 receptor (IL5Ra) and, apart from stopping IL-5 from interacting with its receptor, it also induces antibody-dependent cell-mediated cytotoxicity [13]. The DREAM and MENSA RCTs demonstrated mepolizumab’s ability to reduce the rate of exacerbations and improve lung function and symptom control, while SIRIUS demonstrated a corticosteroid-sparing effect [135,136,137]. RCTs for reslizumab showed a reduction in exacerbation rates and improvement in lung function and asthma control compared to placebo. Reslizumab is administered intravenously and it is less effective with eosinophil counts less than 400/μL [138,139]. Benralizumab showed a significant reduction in annual exacerbation rates, an improvement in symptom control and FEV1, an ability to reduce the use of oral corticosteroids maintaining asthma control and a reduction in asthma exacerbations [140,141,142].

Route of administration, dosing schedule, comorbidities and blood eosinophil cut-offs guide the therapeutic choice between these agents targeting the IL-5 pathway.

Recently, tezepelumab, an anti-TSLP antibody, has been approved by the FDA for the treatment of severe asthma in patients >12 years. In RCTs, tezepelumab reduced the annual rate of exacerbations and improved lung function, asthma control and asthma-related quality of life irrespective of the presence of T2 inflammation biomarkers; however, the greatest benefit on the reduction of exacerbations was observed in patients with high FeNO levels and high blood eosinophil counts.

Hence, tezepelumab seems to be effective in a broader asthma population including those with T2-high and -low endotypes. GINA has suggested a trial with anti-TSLP for patients with T2-low inflammation based on local eligibility criteria, but data are insufficient for patients receiving maintenance OCS [82]. On the other hand, for patients on maintenance oral corticosteroids and with no signs of T2 inflammation, a trial with anti-IL4R can be considered [82].

Many more agents are under development with the aim of further optimizing treatment for asthma patients.

## 7. Conclusions

Allergic asthma is a common phenotype within the diverse spectrum of this chronic disease. A T2-high inflammation profile shapes allergic asthma’s clinical presentation, with allergen exposure provoking relevant symptoms in sensitized patients. Despite its frequent overlapping with other T2-high endotypes, the distinct existence of allergic asthma is beyond doubt. Several biomarkers exist to diagnose allergic asthma and more are under investigation with the aim of proving their value in predicting and monitoring the course of the disease and response to treatment. With allergen immunotherapy being the only etiological therapy for allergic diseases and asthma that works by promoting immunologic tolerance, other tailor-made interventions have risen. Biologics as add-on treatments for severe asthma patients are at the forefront but they still need to be investigated, not only for their effect on controlling asthma inflammation but also for their disease-modifying effect and potential interference with the course of the disease in allergic—but not exclusively—patients. Marching into the era of precision medicine, biomarker-based interventions cut and sewn to each patient are only the first stop in the long yet thrilling journey of the young, dynamically developing field of allergy treatment.

## Figures and Tables

**Figure 1 jpm-12-01162-f001:**
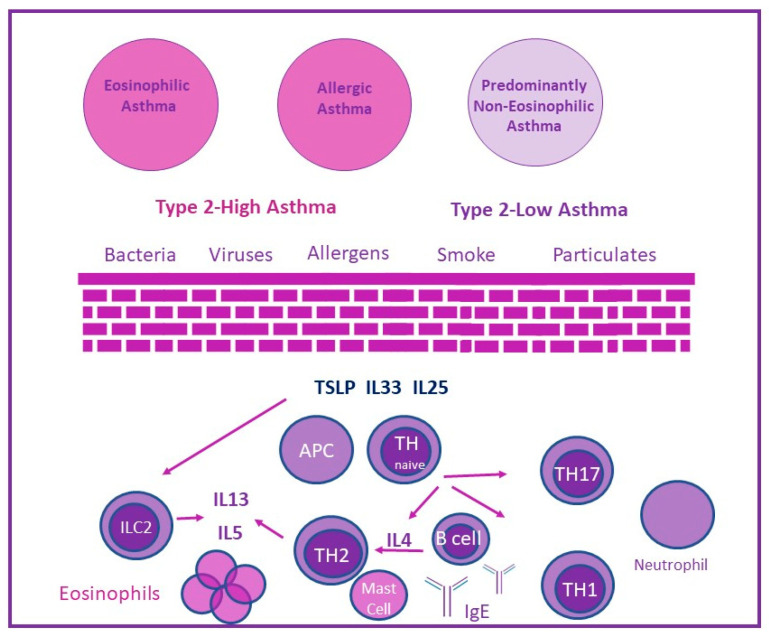
Endotypes of asthma. The activation of the type 2 inflammatory pathway leads to production of IL-4, IL-13 and IL-5 (type 2 cytokines). Allergic asthma is a type 2-high inflammation asthma endotype. Eosinophilic inflammation can be present in the absence of allergic inflammation in the eosinophilic asthma endotype. In type 2-low pathways, activation of TH1 and TH17 cells leads to neutrophilic inflammation (predominantly non-eosinophilic endotype).

**Table 1 jpm-12-01162-t001:** Key points of allergic asthma in the era of personalized medicine.

Key Points
Allergic asthma is the most common phenotype of asthma. It is associated with sensitization to environmental allergens and asthma related symptoms upon exposure.
A T2 inflammatory pathway contributes to allergic asthma mechanisms.
Allergic comorbidities such as allergic rhinitis and atopic dermatitis often coexist with allergic asthma.
Diagnosis of allergic asthma is based on in vivo (skin prick tests) and/or in vitro (allergen specific IgE levels, Component Resolved Diagnosis) documentation of allergic sensitization.
Specific biomarkers such as serum IgE, eosinophils, and epidemiological characteristics (early-onset, atopic comorbidities) contribute to identifying patients with allergic asthma.
Besides inhaled treatments, allergen avoidance, specific immunotherapy, and biologicals represent additional therapeutic options for allergic asthma.
Biologics targeting T2 inflammation exist for management of allergic eosinophilic asthma.

## Data Availability

Not applicable.

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
