# Peer review of "Allergic Asthma in the Era of Personalized Medicine"

_jpm, 2022, doi:10.3390/jpm12071162_

Round 1

Reviewer 1 Report

This review by Papapostolou and Makris summarises certain aspects of allergic asthma, including treatment options. The manuscript in its present form is not well written and contains numerous examples of poor style, grammatical mistakes and poor formatting.

I also have several issues with the content: For example, on line 68 the authors state that allergic asthma…“typically continues into adulthood”. This statement is not properly referenced and I’m not convinced that this is true, since there are many articles describing that asthmatic children are more likely to outgrow their symptoms than not, at least in general terms. The pathogenic mechanisms are also not well described. For example, the authors state that “cross linking of the IgE on the receptors leads to activation of MC and basophils and secretion of the preformed mediators”. This sentence is not only poorly written but only alludes to preformed mediators, without even mentioning what they are. Furthermore, these cells are also capable of producing a variety of newly-generated mediators that perform a pivotal role in driving the symptoms of allergic asthma as well as performing immunomodulatory functions. The final section is rather selective and does not provide a comprehensive overview. Other therapeutics used successfully in allergic asthma therapy are not sufficiently mentioned in any detail (e.g. SABA, LABA, cromoglycate, montelukast, methylxanthines etc.). Some of these sections could also benefit from a figure or table.

Example of poor writing style, formatting issues and spelling/grammatical mistakes:

Line 27: poor style. Avoid using adjectives such as “huge”

Spacing issues throughout the manuscript. E.g. “lying,frequently” on line 36; line 37

Lines 54/55: Poor style. Please re-write the sentence.

Define RCT, RCTs (randomized controlled trial??), LABA, MC etc.

Sentence beginning line 56: Poor/missing use of commas. Sentence too long.

Line 112: correct “…in the asthma inflammation” to “associated with asthma inflammation”.

Line 115: missing definite article, missing hyphen (“   the T2-high endotype”). Similar grammatical mistakes also with line 117…Missing hyphens also common throughout.

Line 129: (TH2).

Line 131, rewrite sentence “Those specific IgE antibodies are bounding on high affinity IgE receptors…”

Line 154: sewed?

Line 174: has been show?

Line 176: corelated (correlated)

Line 273: LAMA?

Line 332: INF-α?

Author Response

Please see attachement.

Reviewer 2 Report

The review is a nice summary of endotypes, mechanisms, biomarkers and the most recent developments in asthma treatment.

There is a need to carefully proofread the document as there are typos and some grammatical issues. I think the article would benefit from a diagram that provides a visual representation of the various sub-categories/endotypes.

Some  specific comments:

1. Large number of abbreviations that are not always defined and/or difficult to keep up with. Please check through carefully and consider a table of abbreviations.

2. On page 4 you mention a link between IgE ad mucus hyper secretion but do not reference a specific study to show there is evidence for this on a mechanistic level. You also state that ILC2s are mainly important in T2, non-allergic endotype - but do not provide evidence of this.

3. A diagram or table to support reader in drawing key pieces of information.
